# Microwave Radiation as a Pre-Treatment for Standard and Innovative Fragmentation Techniques in Concrete Recycling

**DOI:** 10.3390/ma12030488

**Published:** 2019-02-05

**Authors:** Maarten Everaert, Raphael Stein, Simon Michaux, Vincent Goovaerts, Carlo Groffils, Simon Delvoie, Zengfeng Zhao, Ruben Snellings, Peter Nielsen, Kris Broos

**Affiliations:** 1Flemish Institute for Technological Research (VITO), Unit Sustainable Materials Management, Boeretang 200, 2400 Mol, Belgium; ruben.snellings@vito.be (R.S.); peter.nielsen@vito.be (P.N.); kris.broos@vito.be (K.B.); 2Department of Processing and Recycling, RWTH Aachen University, Wüllnerstraße 2, 52062 Aachen, Germany; raphael.stein@iar.rwth-aachen.de; 3Geological Survey of Finland, P.O. Box 96, Betonimiehenkuja 4, F1-02151 Espoo, Finland; simon.michaux@gtk.fi; 4Department of Architecture, Geology, Environment and Constructions, University of Liège, Quartier Polytech 1, Allée de la découverte 9, 4000 Liège, Belgium; s.delvoie@uliege.be (S.D.); zengfeng.zhao@uliege.be (Z.Z.); 5MEAM Microwave Test Center, Industrieweg 1119, 3540 Herk-de-Stad, Belgium; vincent.goovaerts@meam.be (V.G.); carlo.groffils@meam.be (C.G.)

**Keywords:** aggregates, cement matrix, concrete, recycling, electrodynamic fragmentation, microwave weakening, crushing

## Abstract

Recent advances in concrete recycling technology focus on novel fragmentation techniques to obtain aggregate fractions with low cement matrix content. This study assesses the aggregate liberation effectiveness of four different treatment processes including standard and innovative concrete fragmentation techniques. Lab-made concrete samples were subjected to either standard mechanical crushing technique (SMT) or electrodynamic fragmentation (EDF). For both fragmentation processes, the influence of a microwave weakening pre-treatment technique (MWT) was investigated. A detailed analysis of the particle size distribution was carried out on samples after fragmentation. The >5.6 mm fraction was more deeply characterized for aggregate selective liberation (manual classification to separate liberated aggregates) and for cement matrix content (thermogravimetric measurements). Results highlight that EDF treatment is more effective than SMT treatment to selectively liberate aggregates and to decrease the cement matrix content of the >5.6 mm fraction. EDF fully liberates up to 37 wt.% of the >5.6 mm natural aggregates, while SMT only liberates 14–16 wt.%. MWT pre-treatment positively affects aggregate liberation and cement matrix removal only if used in combination with SMT; no significant effect in combination with EDF was recorded. These results of this study can provide insights to successfully implement innovative technology in concrete recycling plants.

## 1. Introduction

Continuous global population growth and ongoing economic development demand a thriving construction industry. This industry strongly depends on the production of concrete, the most widely used building material. Global production of concrete has increased up to 25 Gt per year [1]. Yet, the amount of demolition concrete waste has increased over the past decades. In response, recycling practices for concrete waste have emerged due to environmental and economic concerns.

Flanders (Belgium) has a long history in recycling and reuse of concrete and aggregates, especially since Flemish legislation banned landfilling of construction and demolition waste in 2009 [2]. In 2015, up to 14.6 Mt of recycled aggregates were recovered from construction and demolition waste in Flanders, which accounts for 50% of the aggregates demand in Flanders [3]. Although the recycling rate of concrete in Flanders increases year after year, current recycling practices yield low-grade recycled aggregates and are therefore responsible for the “downcycling” of the concrete constituents. In 2015, only 32% of the recycled concrete aggregates were used in the production of new concrete, such as precast products, ready-mix concrete and stabilized sand [3]. This is due to the fact that, conventionally, concrete fragmentation occurs via mechanical crushing and/or grinding, e.g., using impact crushing or jaw crushing [4]. The quality of these recycled aggregates is often suboptimal in view of replacing a large fraction of natural aggregates in new structural or non-structural concrete [5,6]. A high content of residual cement matrix in the recycled aggregate fraction results in a high porosity and increased water absorption [7,8], which negatively affects workability, strength and durability of concrete prepared from these aggregates [9,10,11]. As a result, low-grade recycled aggregates produced in Flanders and obtained with current fragmentation technologies, are mostly used in applications with less strict aggregate requirements, e.g., primary foundation and levelling works [3]. In the future, however, the supply of low-grade aggregates is likely to grow more rapidly than the demand, thereby increasing the pressure on recycling technologies yielding high-grade recycled aggregates that can be reused in new structural and non-structural concrete. High-grade recycling is therefore an important step towards a circular building industry.

Several new concrete fragmentation techniques have recently been suggested to produce cleaner recycled aggregate fractions, including: i) microwave-assisted fragmentation (MWT) and ii) electrodynamic fragmentation (EDF). The technique of microwave-assisted fragmentation uses microwaves to internally heat concrete waste material. The moisture content of the different concrete constituents is strongly linked to the absorption of the applied microwave energy, because it directly affects the dielectric properties of the constituents [12]. As a result of the microwave heating, high differential stresses are generated in concrete, especially at the interface with the embedded aggregates [13]. Consequently, concrete can delaminate and disintegrate into smaller fractions. Electrodynamic fragmentation is based on the discharge of electrical pulses throughout waste concrete material immersed in water. When subjected to a high electric field, concrete constituents polarize on a magnitude and a location dependent on their intrinsic electrical properties. Generally, this imbalance of electrical charges is maximized at the aggregate boundaries, where local discharges are accompanied by thermal expansion. This can result in a radial shockwave, causing fragmentation of concrete [14]. Both microwave-assisted and electrodynamic fragmentation techniques aim to be more selective in the removal of the hardened cement matrix from the aggregates compared to standard mechanical fragmentation techniques.

For recent years, microwave-assisted fragmentation and electrodynamic fragmentation have been assessed for their effectiveness of aggregate selective liberation [13,15,16,17,18]. Touzé et al. [14] proved the robustness of electrodynamic fragmentation in liberating natural aggregates with varying properties from concrete. Bru et al. [15] suggested to use a microwave pre-treatment prior to mechanical fragmentation to increase concrete fragmentation and aggregate liberation. Both studies pointed out the promising capabilities of microwave and electrodynamic techniques in order to remove natural aggregates from concrete. However, a combination of different novel techniques, e.g., a microwave pre-treatment followed by electrodynamic fragmentation has not been tried before and holds promise to further improve selective aggregate liberation from concrete waste.

In this study, this concept was tested by assessing the combined effect of a microwave pre-treatment prior to electrodynamic fragmentation or standard mechanical treatment (impact crusher), on concrete fragmentation and aggregate liberation. The obtained fragmented material from the different treatments was evaluated based on the obtained particle size distribution. Additionally, the >5.6 mm fraction of the obtained material was characterised based on aggregate selective liberation (classification) and cement content.

## 2. Materials and Methods

### 2.1. Materials

Lab-made concrete was used in this study, prepared according to the standard EN 206-1 [19]. The aggregate material used for this concrete was Moselle gravel (4/8 mm), consisting of sandstone, quartzite and slate, with a maximum grain diameter of 8 mm. The cement was Portland cement CEM I 42.5 R and the sand was Rhine sand (0/4 mm). The particle size distribution of the natural aggregates used for the preparation of the concrete samples was determined according to standard EN 933-1 [20]. The aggregate sample was screened at 0.25, 0.5, 1.0, 2.0, 4.0, 5.6, 8.0, 10.0, 16.0 mm and weighted in order to construct the particle size distribution. The constituents used for the casting of the concrete consists of 370 kg m^−3^ cement, 249 kg m^−3^ water and 1940 kg m^−3^ aggregates (of which 1098 kg m^−3^ Rhine sand and 842 kg m^−3^ Moselle gravel), thereby also meeting the standard guideline of a water/cement ratio of 0.55. Additional water (38 kg m^−3^) was added to account for water absorption by aggregates. Afterwards, the prepared concrete was cured for one year (Figure 1). Smaller concrete blocks were prepared by cutting the concrete into samples with dimensions of 50 mm × 50 mm × 20 mm. The cured blocks, 24 in total, had an average weight of 110 g and belonged to the C25/30 concrete strength class.

### 2.2. Fragmentation Methods

In order to fragment the concrete samples and assess aggregate selective liberation by several treatments, three different techniques were used: i) microwave treatment (MWT), ii) electrodynamic fragmentation (EDF) and iii) standard mechanical treatment (SMT). For MWT, a monomode microwave MEAM Mono-Explorer 6 (Herk-de-Stad, Belgium) was used. Without a pre-drying procedure, the samples were exposed to microwave radiation with a power of 3 kW and frequency of 2.45 GHz during a 60 s period (including the 5 s acceleration period). This MWT had an estimated non-optimised energy consumption of 450 kWh t^-1^. During the MWT, the concrete samples were placed on two Teflon blocks for an optimal exposure. The EDF treatment was performed using a Selfrag lab unit (Kerzers, Switzerland), which operated at 180 kV and applied 30 pulses at a frequency of 5 Hz. The spacing between both electrodes corresponded to 40 mm. The estimated energy consumption of EDF was 30 kWh t^−1^. The SMT used a Hazemag impact crusher (Dülmen, Germany). The crusher operated with 1.1 kW power and with a rotation speed of 1600 rpm (~20 m s^−1^), resulting in an energy consumption around 0.5 kWh t^−1^. This mechanical treatment was included as a standard technique for concrete fragmentation.

### 2.3. Experimental Design

The different fragmentation treatments explored and evaluated in this study are presented in Figure 2. First, a selection of the concrete samples was exposed to MWT. The weight of the concrete samples was determined before and after this treatment. Using a Fluke visual IR thermometer (type VT04), the surface temperature of the samples after this treatment was measured, which reached values up to 250 °C. A visual examination of MWT samples showed the formation of cracks in several concrete blocks (Figure 3), yet no fragmentation of the concrete samples occurred. Subsequently, non-treated concrete samples as well as MWT samples were exposed to either SMT or EDF. Samples obtained from SMT were screened at 8 mm. The overflowing fraction of this procedure (>8 mm) was crushed a second time, in order to mimic the loop feed in a large-scale processing plant. Taken together, a total of four different fragmentation treatments were compared in this study. For each of these procedures, six replicates were included.

### 2.4. Characterization Methods

Samples obtained from the different fragmentation treatments were characterised based on size distribution, aggregate selective liberation (classification) and cement matrix content. First, the particle size distribution of the obtained fragmented samples was determined. Samples were screened at 0.25, 0.5, 1.0, 2.0, 4.0, 5.6, 8.0, 10.0, 16.0 mm (same sieves as were used for the natural aggregates). Gravimetric measurement of each fraction was used to obtain the particle size distribution curve. Secondly, a classification step was performed on the >5.6 mm fraction of each sample obtained after concrete fragmentation. This fraction was sorted by hand into three particle classes based on visual particle characteristics: fully liberated aggregates (FL), aggregates with adhering cement matrix (AC) and chunks of cement matrix (CM). The FL class had no visual signs of the presence of cement matrix on the aggregate and the AC class had pieces of cement matrix adhering to single aggregates. The CM class included (generally larger) pieces of cement matrix, which could also contain several aggregates. The focus on the >5.6 mm fraction in this study was merely for the practical considerations related to the hand-sorting. Thirdly, the cement content of the obtained particle classes was measured using the thermogravimetric method. This method is based on the hydration reaction of cement with water during aging, assuming that (i) cement on average reacts with water for 24% of its weight [21] and (ii) this hydration reaction has been completed after curing the concrete for one year. The free water content (*w*_0_) in each sample was determined from the weight loss by a thermal treatment at 105 °C for 24 h. Consequently, the chemically bound water content (*w_t_*) of these samples was determined by the additional weight loss after a thermal treatment at 550 °C for 2 h, causing further dehydration and dehydroxylation of cement hydration products [22]. The loss of chemically bound water during the MWT pre-treatment is assumed to be negligible compared to the total chemically bound water in the samples and therefore not taken into account in this analysis. From the obtained *w*_0_ and *w_t_* values, the cement content (*c*) in each sample was estimated using the following formula:
(1)$$c=\frac{\left(w_{0}-w_{t}\right)\times 100}{0.24}$$

To determine the cement content of the particle classes, the FL, AC and CM particles were hand-sorted from the >5.6 mm fraction. The average cement content of each particle class was multiplied with the fraction of the corresponding class in each of the >5.6 mm samples, in order to obtain the total cement content of each >5.6 mm sample.

### 2.5. Statistical Analysis

The statistical significance of the differences in size fraction, aggregate selective liberation, aggregate recovery and cement content between the different fragmentation treatments was examined via a paired student t-test, thereby separating the treatments with or without MWT; the significance of a MWT pre-treatment was tested via a one-way ANOVA analysis. For both, the statistical program JMP Pro 11 (SAS) was used.

## 3. Results

### 3.1. Particle Size Distribution

The particle size distributions of the natural aggregate and of the samples obtained after fragmentation treatments are presented in Figure 4. The particles obtained from the SMT treatments are coarser compared to those obtained from the EDF treatments. Generally, the effect of the MWT pre-treatment on the particle size distribution of the obtained material is limited. The largest difference as result of the MWT is observed for the particle fractions obtained from SMT, showing the MWT-SMT sample to be slightly finer compared to the SMT sample for the fraction between 2 and 8 mm.

### 3.2. Liberation of Aggregates

The >5.6 mm fraction from each fragmentation treatment, also indicated in Figure 4, is presented in Figure 5a. The MWT pre-treatment reduces the mass of the >5.6 mm fraction obtained after SMT, while this pre-treatment has no significant effect on the mass of the >5.6 mm fraction obtained after EDF. Furthermore, from the concrete composition it is calculated that the mass of the >5.6 mm natural aggregates in the concrete mix relative to the total mass of the concrete mix is 16.4% of the total sample mass, as marked in Figure 5a. Since the mass of the >5.6 mm fraction of the treated concrete material relative to the total sample mass is well below this value, this means a significant part of the natural aggregates disintegrated during SMT and EDF.

The relative amount of FL aggregates in the >5.6 mm fraction is determined to assess the aggregates liberation during the different treatments (Figure 5b). The amount of fully liberated aggregates is much higher for EDF compared to SMT. Up to 78% of the mass of the >5.6 mm fraction from EDF can be attributed to recovered FL aggregates. A significant effect of MWT on the aggregate selective liberation is present for SMT, which is also shown in the comparison of the obtained hand-sorted particle classes (Figure 6). This effect is, however, not significant for EDF. Next to the amount of FL aggregates, also the relative content of AC and MC is determined (Table 1).

From the mass of the FL aggregates in the >5.6 mm fraction and the initial concrete aggregate content, the recovery of FL aggregates with the different treatments is calculated (Figure 5c). The highest values for aggregate recovery are obtained for EDF (37%) and for MWT-EDF (35%). SMT has the lowest recoveries, that is 14% for SMT and 16% for MWT-SMT.

### 3.3. Cement Content

Using thermogravimetric analysis, the cement content of the particle classes present in the >5.6 mm fraction is determined (Figure 7). In general, this figure shows that the particle classes, obtained from different fragmentation treatments, have a very similar cement content between the different treatments, which proves hand-sorting of the recovered particles to be a robust and reliable method. Therefore, the average cement content of the particle classes are determined for the different fragmentation treatments (Table 2). The particle classes obtained from different fragmentation treatments have a very similar cement content, yet the relative amount of these particle classes can differ a lot between the samples (Figure 5b). The FL particles have a low cement content, on average 2.7%. With an increasing visual presence of cement matrix on the hand-sorted particles, also the cement content increases. This is illustrated by the higher average cement content of AC particles (5.9%) compared to FL particles. Finally the CM particles, selected on a very pronounced presence of cement matrix, have the highest average cement content of the three particle classes (13.1%). This means that the relative cement content of a fraction obtained from the different fragmentation treatments can be estimated (Figure 8).

The cement content of the >5.6 mm fractions is determined by the relative content of the FL, AC and CM classes in the >5.6 mm fraction (Table 1) combined with the average cement content of each class (Table 2). The obtained data show a clear inverse relation between the amount of fully liberated aggregates in a >5.6 mm fraction (Figure 5, middle) and the average cement content of that fraction (Figure 8). This means that samples from both EDF treatments are significantly lower in cement content compared to samples from the SMT treatments. The MWT pre-treatment slightly reduces the cement content of the SMT treatment, that is from 9.4% to 8.1%, while the MWT pre-treatment prior to EDF treatment has no significant effect.

## 4. Discussion

### 4.1. Liberation of Aggregates

Overall, the data show that EDF, followed by a >5.6 mm size separation, is able to deliver a cleaner recycled aggregate fraction in comparison with the other tested treatments. Only in combination with SMT, MWT slightly increases the liberation and recovery of aggregates; no significant effect is shown for MWT as pre-treatment for EDF. Previous studies also confirmed the effectiveness of a MWT pre-treatment prior to SMT [13,15,23]. It is hypothesized that the significance of the MWT pre-treatment could be linked to the mechanisms of concrete fragmentation by SMT and EDF. The fragmentation mechanism of each of the techniques strongly determines the aggregate selective liberation in the obtained sample. On the one hand both MWT and EDF specifically target the interface between the aggregates and the cement matrix. On the other hand SMT is less selective in the liberation of the aggregates. It is, therefore, likely that the microwave pre-treatment can have an additional effect if followed by a less specific fragmentation, such as SMT. It can be noted that the procedure used in this study to assess aggregate liberation effectiveness is based on a screening method, with limited focus on optimisation. Therefore, the relative differences between the different treatments are of main importance, as the absolute values for liberation effectiveness could possibly be further improved.

### 4.2. Cement Content

For the >5.6 mm fractions obtained from the different treatments, a link between the cement content of a recycled aggregate fraction and the relative abundance of the particle classes (FL, AC, CM) in that fraction is expected. In this study, the total cement content of a >5.6 mm fraction clearly decreases with the increasing liberation of aggregates in that fraction. The FL particles have the lowest cement content, on average 2.7%. Remaining cement of these FL aggregates is likely related to minor amounts of cement matrix remaining in surface imperfections of the aggregates. The CM particles have the highest average cement content of the three particle classes (13.1%), which is comparable to the cement content of the initial concrete samples (14.2%). The results of the cement content of the material obtained from the different treatments are in line with the results from the aggregate liberation. The lowest cement content is obtained for EDF and only for SMT a MWT pre-treatment results in a significant decrease of the cement content. Here, it can be noted that lowering the cement content with aggregate liberation is only possible if the attached cement matrix not only is removed from the aggregate surface during treatment but also is excluded from the >5.6 mm fraction. In other words, not only the complete liberation of aggregates is a determining factor to decrease the cement matrix content in a certain fraction of fragmented concrete but also the impact of the different treatments on the fragmentation of the cement matrix phase plays a role.

### 4.3. Feasibility of Using MWT- and EDF-Technology in Concrete Recycling Systems

This study used lab-made concrete blocks with the strength class of C25/30. In comparison with many other concrete formulations, the compressive strength of these concrete samples is relatively low, which is related to the high water/cement ratio used for the concrete samples [24]. It is expected that the strength of the concrete, resulting from the hardness of the matrix, will influence the relative aggregate selective liberation effectiveness of the different treatments examined in this study. The water/cement ratio is an important factor for effectiveness of the proposed treatments: a lower water/cement ratio of the initial concrete would increase the compressive strength of the concrete due to a strength increase of the cement matrix. In such a concrete, the strength of the cement matrix and the natural aggregates would be more comparable, which has implications for fragmentation and liberation. In that case, it is expected that aggregate liberation via non-selective techniques, such as the SMT in this study, would be harder. Gains in aggregate liberation associated to an optimisation of the operating parameters of mechanical treatment will likely be limited. Techniques working more specifically on the aggregate-matrix interface, such as microwave treatment or electrodynamic fragmentation, can also be affected by a decrease of the water/cement ratio, e.g., a lower energy absorption by concrete during microwave treatment causing limited impact. Future research should be conducted in order to determine the relative liberation effectiveness of standard and innovative fragmentation techniques for different concrete materials with varying water/cement ratios.

It is implied that, although a significant effect was obtained for the MWT pre-treatment on the effectiveness of SMT, the EDF technique remains most promising for aggregate selective liberation from concrete: a recycled aggregate fraction with high aggregate selective liberation and low cement matrix content was obtained with EDF. However, so far the technology of EDF has not yet been applied in concrete fragmentation systems at large scale and current concrete fragmentation installations remain mainly based on mechanical fragmentation. Therefore, an additional microwave based pre-treatment to the existing fragmentation technology is likely a more feasible approach to increase aggregate liberation on short-term basis, even when the gains in aggregate liberation will be smaller.

Next to the aggregate selective liberation of the different treatments, the energy consumption of these treatments is a major issue. Menard et al. [16] found that a lab-scale EDF treatment is more energy efficient in comparison with a lab-scale MWT treatment and Touzé et al. [14] reported lab-scale EDF treatment to be slightly more energy efficient in comparison with conventional crushing. However, it can be noted that lab-scale fragmentation equipment, such as the instruments used in this study, is typically not designed for an energy efficient performance. For example, Bru et al. [25] illustrated that the specific energy consumption of a batch lab-scale EDF system during fragmentation largely exceeded that of a continuous pilot-scale EDF system. Therefore, it is clear that the feasibility of the use of EDF and MWT techniques in large-scale treatment of concrete waste strongly depends on the ongoing development of new MWT- and/or EDF-based systems.

## 5. Conclusions

A high aggregate selective liberation during a concrete fragmentation treatment is of key importance to recover the aggregate fractions for high-quality applications. Therefore, research focusses on novel technology and treatments to increase aggregate selective liberation and thus aggregate quality. From this approach, this study assesses the significance of microwave weakening pre-treatment on the aggregate selective liberation from lab-made concrete samples. Results show that EDF treatments are more effective in aggregate selective liberation compared to SMT. Indeed, EDF resulted in a >5.6 mm recycled aggregate fraction with more FL aggregates and an overall lower cement content. The effect of a microwave pre-treatment was only significant in terms of aggregate liberation and lowering the cement content if followed by SMT, while MWT did not significantly influence EDF recycled aggregates. Future research can further elaborate on the effect of MWT pre-treatment for SMT or EDF treatment on concrete with varying strength properties and water/cement ratios, as this is likely to affect the relative effectiveness of the liberation treatments tested here. Also the liberation of aggregates smaller than 5.6 mm obtained via the studied treatments can be assessed.

## Figures and Tables

**Figure 1 materials-12-00488-f001:**
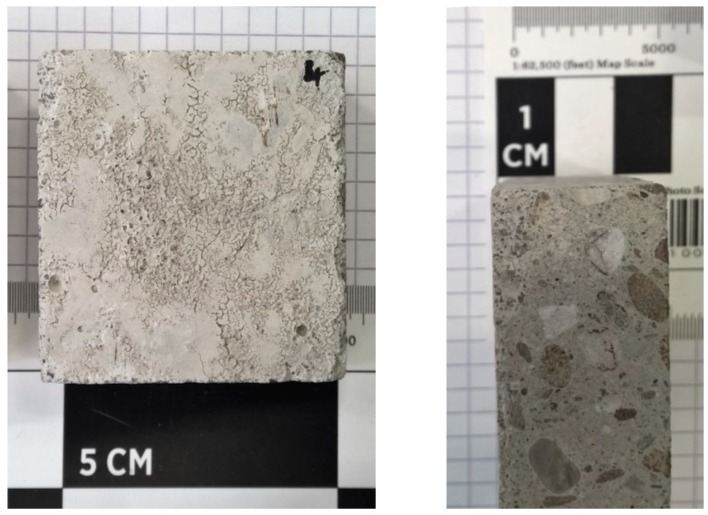
The lab-made concrete samples.

**Figure 2 materials-12-00488-f002:**
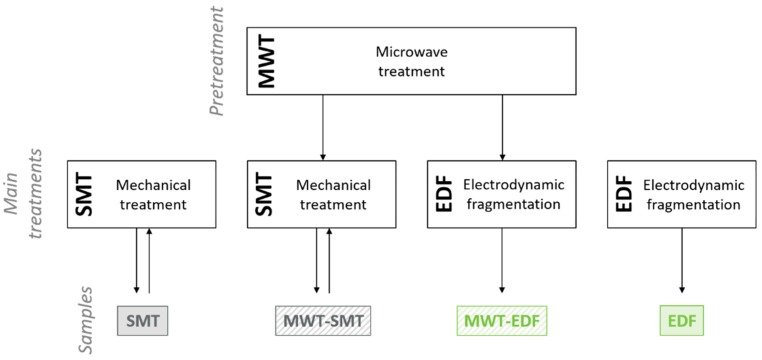
Experimental testing procedure based on microwave pre-treatment (MWT), standard mechanical treatment (SMT) and electrodynamic fragmentation (EDF) techniques, yielding four different fragmented samples.

**Figure 3 materials-12-00488-f003:**
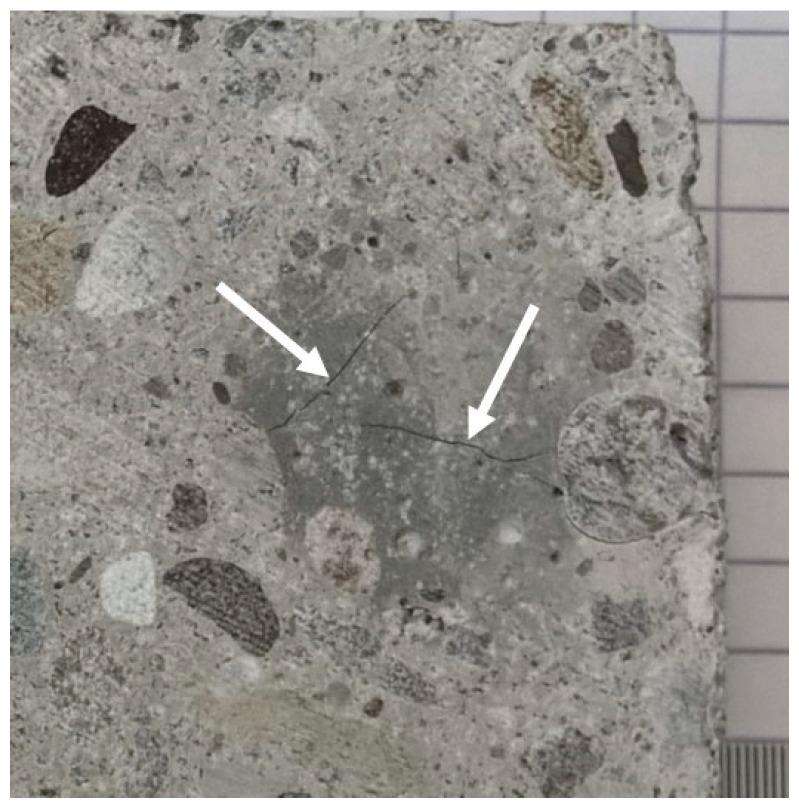
Visible cracks in the concrete blocks, a direct result of the exposure to the MWT treatment.

**Figure 4 materials-12-00488-f004:**
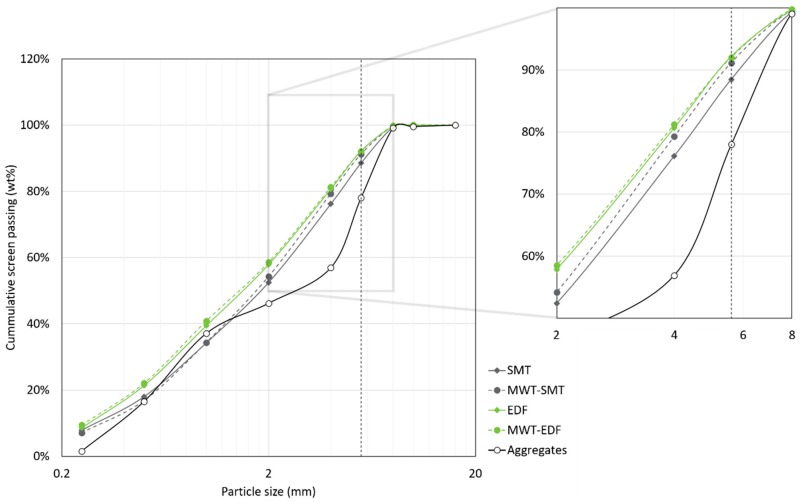
Particle size distribution of the natural aggregates and of the samples obtained after fragmentation using different treatments. Notice the deviation in the distribution between the SMT and the MWT-SMT samples in the 2–8 mm range (inset). The vertical dotted line indicates the >5.6 mm fraction used for further characterization.

**Figure 5 materials-12-00488-f005:**
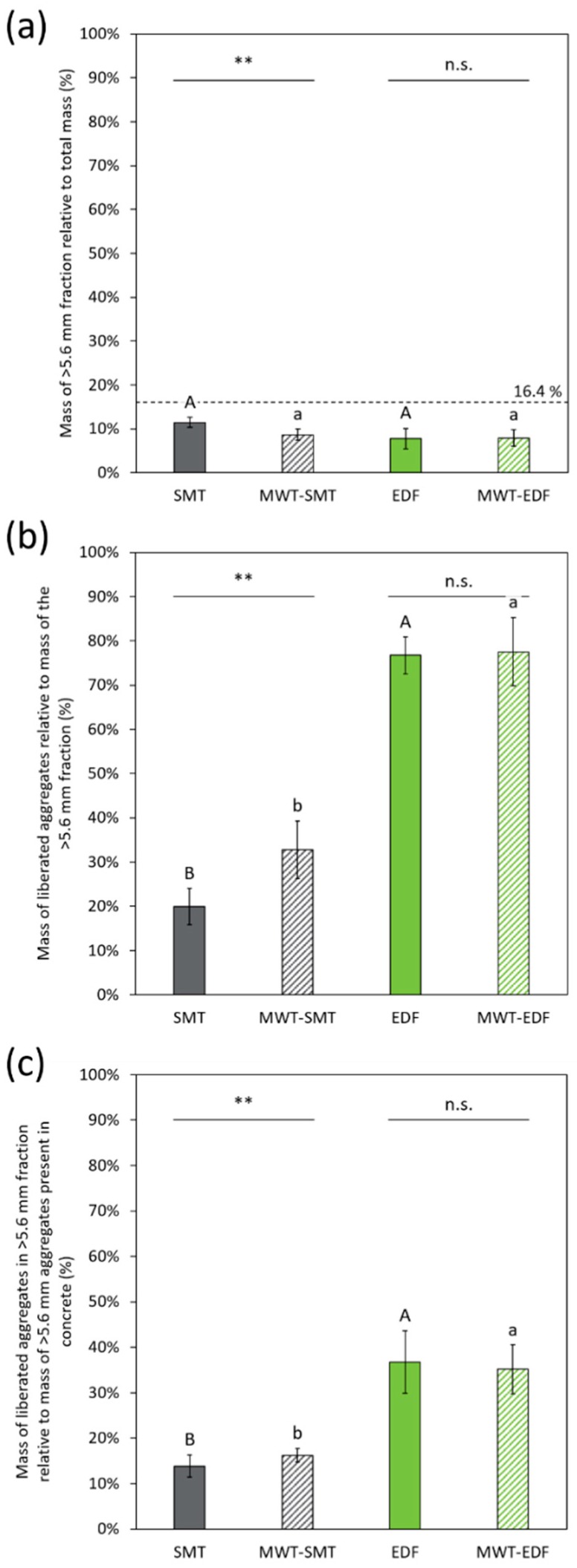
(**a**) The mass of the >5.6 mm fraction relative to the total mass (%), derived from the particle size distributions, presented for each treatment. The dotted line at 16.7% is the aggregate content of the initial concrete; (**b**) The mass of the liberated aggregates relative to the mass of the >5.6 mm fraction (%); (**c**) The liberated aggregates in the >5.6 mm fraction relative to the aggregates present in the concrete (%). The different letters in these graphs indicate statistical (P ≤ 0.05) differences between the treatments without MWT (capital letters) and between the treatments with MWT (lowercase). Significant differences of the effect of MWT on SMT and EDF (six replicates) are indicated by asterisks above the pair of bars (* P ≤ 0.05; ** P ≤ 0.01; ns: not significant). Error bars in the figures represent standard deviation.

**Figure 6 materials-12-00488-f006:**
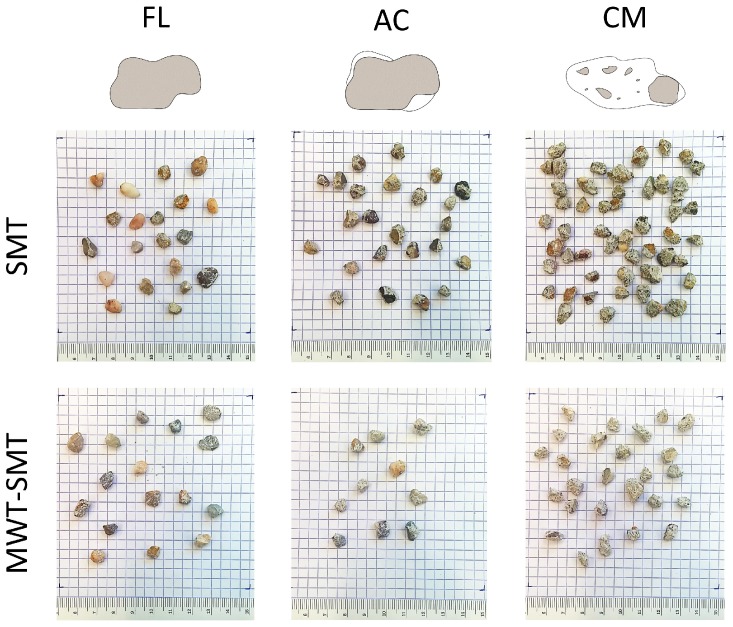
Pictures from the hand-sorted particles of the >5.6 mm fraction obtained after the SMT and MWT-SMT fragmentation treatments, classified as fully liberated aggregates (FL), aggregates with adhering cement matrix (AC) and chunks of cement matrix (CM). This overview shows visually that the CM class has a relative higher contribution for material recovered after SMT compared to material recovered after MWT-SMT. The relative distribution of the recovered aggregates along these classes is in alignment with the results shown in Figure 5b. The measured total mass of the particle classes from each of the treatments is presented in Table 1.

**Figure 7 materials-12-00488-f007:**
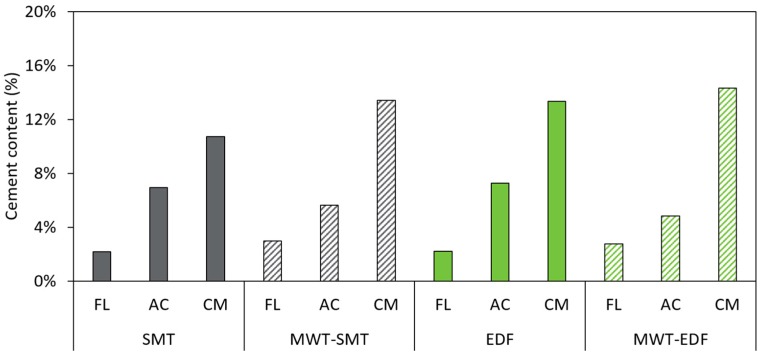
The cement content of the different particle classes (FL: fully liberated aggregates; AC: aggregates with adhering cement matrix; CM: cement matrix chunks) in the >5.6 mm fraction, presented for the different fragmentation treatments.

**Figure 8 materials-12-00488-f008:**
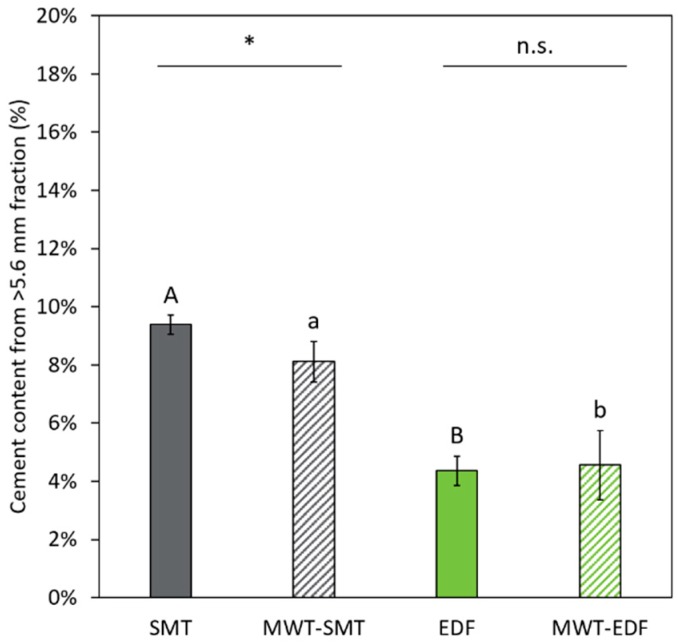
The cement content of the >5.6 mm fraction, derived from the particle class distribution (Table 1) and the particle class cement content. The different letters in these graphs indicate statistical (P ≤ 0.05) differences between the treatments without MWT (capital letters) and between the treatments with MWT (lowercase). Significant differences of the effect of MWT on SMT and EDF (three replicates) are indicated by asterisks above the pair of bars (* P ≤ 0.05; ns: not significant). Error bars in the figure represent standard deviation.

**Table 1 materials-12-00488-t001:** The relative mass content of the fully liberated aggregates (FL), the aggregates with adhering cement matrix (AC) and the chunks of cement matrix (CM), present in the >5.6 mm fraction.

Particle Class	SMT	MWT-SMT	EDF	MWT-EDF
FL	20% ^B^	33% ^b^	77% ^A^	78% ^a^
AC	23% ^A^	22% ^a^	11% ^B^	7% ^b^
CM	57% ^A^	45% ^a^	13% ^B^	16% ^b^

^A,B,a,b^: The different letters indicate statistical (P ≤ 0.05) differences between the treatments without MWT (capital letters) and between the treatments with MWT (lowercase). Six replicates were used.

**Table 2 materials-12-00488-t002:** The average cement content of the different particle classes for the >5.6 mm fraction. The standard deviation is given for each value.

Fraction	FL	AC	CM
>5.6 mm	2.7% ± 0.4%	5.9% ± 0.8%	13.1% ± 2.0%

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
