# Peer review of "Microwave Radiation as a Pre-Treatment for Standard and Innovative Fragmentation Techniques in Concrete Recycling"

_materials, 2019, doi:10.3390/ma12030488_

Round 1

Reviewer 1 Report

This article aims to assess the aggregate liberation effectiveness of four different recycling processes including innovative processes such as microwave pretreatment technique and electrodynamic fragmentation. In particular, this paper aims to study the combination of the microwave pretreatment technique (MWT) followed by the electrodynamic fragmentation (EDF). The > 5.6 mm fraction was more deeply characterized for aggregate selective liberation and for cement matrix content.

Actually, even if it could be interesting to study the combination of two innovative techniques which have showed a potential for concrete recycling, combining the MWT to the EDF could be questionable since both techniques aim first to weaken the material by generating cracks at the grain boundaries (when a low amount of energy is used) and then to break it up when the amount of energy used is increased. From my point of view, the originality is more related to the approach consisting in considering a classification process as a step for separating the coarse liberated aggregates avoiding then the implementation of a concentration process.

Main comments regarding this article are given below:

-       The use of the classification process as a method to separate the liberated aggregates should be mentioned in the abstract.

-       The liberation of the aggregates smaller than 5.6 mm which make up for most of the initial aggregates should be considered or at least mentioned in the conclusion for future works.

Specific comments are given below

-       Section 2.2 Fragmentation methods:

o    The energy used (in kWh/t) for the MWT and EDF treatments of the samples should be mentioned, at least the one consumed at the plug and/or the one injected to the samples if available.

o    The reasons motivating the study of only one set of operating conditions for the SMT, MWT and EDF treatments should be given. In particular, the impact crushing was performed at a very high rotation speed and this may not be favorable for the liberation of the coarse aggregates.

-       Section 2.3 Experimental design:

o    The second treatment of the oversize at 8 mm should be mentioned in Figure 2.

-       Section 2.4 Characterization methods

o    The method used to estimate the cement content is based on the hydration reaction of cement with water during aging, according to the literature. It would have been useful to show that this method was validated for the studied concrete sample and in particular for the MWT treated sample which could have lost some water during MWT treatment (the surface temperature was measured to 250°C but inside it could be much higher).

-       Section 3.1 Particle size distribution

o    It could be useful to indicate if the MWT treatment led to a breakage of the sample and then to a specific PSD before SMT.

o    Line 185-187: It is written that the MWT-SMT sample has a PSD which is slightly finer compared to the SMT sample for the fraction between 2 and 8 mm. However, this is hardly noticeable in Figure 4. Including standard deviations in Figure 4 could be useful for the analysis of the results.

-       Section 3.2 Liberation of aggregates

o    It is written Line 200-201 that a significant part of the natural aggregates disintegrated during SMT and EDF. It would then have been interesting to investigate other operating conditions in order to study if some of them could lead to a lower disintegration of the natural aggregates. Or at least to mention that better results may be obtained at other operating conditions.

o    Figure 5: MWR Line 216 should be replaced by MWT.

o    Figure 5: it could be useful to show the standard deviations.

-       Section 3.3 Cement content

o    Figure 8: MWR Line 263 should be replaced by MWT.

o    Figure 8: it could be useful to show the standard deviations.

-       Section 4.2 Cement content

o    Line 293: The reference n°16 seems not correct for the related statement.

Author Response

The reply to comments can be found in the attached pdf file.

Reviewer 2 Report

This paper is in good order with good quality.

The authors commented on the feasibility of the method in chapter 4.3. However, there is major concern on the selected water to cement ratio. The w/c used in the lab-made concrete was 0.55, which was way too high for most particle concrete. The only reason I can think of why such a high w/c was selected was it was much more efficient and effective in the microwave treatment.  And I’d assume it was for the same reason, a 28-day age was selected for the lab-made concrete, which was also significantly shorter than most demolished concrete.

That being said, it was too late to change the whole experiment design, and I would still recommend this work to be published. I’d suggest the authors add more explanation in Section 4.3, discussing how w/c and age could affect the efficiency microwave treatment and how will it affect the feasibility of the proposed method.

Author Response

(The authors gave the same response as above.)

Reviewer 3 Report

I consider that the paper is worth to be published in its current form, I did not find any conceptual or methodological gap. Spelling and overall format also seems to be correct.

References are actual, and I really enjoyed the reading. It is well written and relatively easy to follow and understand.

Best regards,

The Reviewer

Author Response

The authors thank the reviewer for the time and the effort they put into the manuscript.

Round 2

Reviewer 1 Report

The authors made response to the review comments and improved the manuscript.

However, I still have some comments regarding this updated manuscript.

Reply to Comment n°3 about the energy.

It is true that the lab-scale equipment are not as energy efficient than pilot-scale or industrial scale equipment. However, it still gives some information about the energy range requested by the process.

Two comments are related to this aspect:

     o    The energy consumption of the microwave treatment was estimated to be 450 kWh/t. The monomode microwave treatment is usually quite energy efficient so it is quite surprising that no fragmentation of the concrete samples occurred during this treatment. In all cases, these operating conditions should not be considered as “relatively mild operating conditions” as written line 138.

   o    The estimated energy consumption of the impact crusher (i.e. 420 kWh/t) seems extremely high for a crushing step. If not measured during this study, it may be better to indicate the energy usually requested for this kind of step. 

It is written line 336 that “Techniques working more specifically on the aggregate-matrix interface, such as microwave treatment or electrodynamic fragmentation, will likely be less affected by an increase in the strength of the cement matrix” i.e. by a decrease in the water/cement ratio (since it is written line 330 that “a lower water/cement ratio of the initial concrete would increase the compressive strength of the concrete due to a strength increase of the cement matrix.” However, previous works on the use of microwave for concrete recycling showed that water plays an important role in the liberation of aggregate during this process. So this sentence should be updated.

Author Response

Reply to comments in attached pdf file
